# Concomitant Sub-Chronic Administration of Small-Size Gold Nanoparticles Aggravates Doxorubicin-Induced Liver Oxidative and Inflammatory Damage, Hyperlipidemia, and Hepatic Steatosis

**DOI:** 10.3390/molecules28020796

**Published:** 2023-01-13

**Authors:** Ghedeir M. Alshammari, Mohamed Anwar Abdelhalim, Mohammed S. Al-Ayed, Laila Naif Al-Harbi, Mohammed Abdo Yahya

**Affiliations:** 1Department of Food Science & Nutrition, College of Food and Agricultural Sciences, King Saud University, Riyadh 11451, Saudi Arabia; 2Department of Physics and Astronomy, College of Sciences, King Saud University, Riyadh 11451, Saudi Arabia

**Keywords:** gold nanoparticles, doxorubicin, hepatic steatosis, SREBP1, NF-κB, Nrf2

## Abstract

This study examined the effect of gold nanoparticles (AuNPs) on doxorubicin (DOX)-induced liver damage and steatosis in rats and tested its effect mechanism. Wistar male rats were divided into four groups (each of eight rats) as control, AuNPs (50 µL of 10 nm), DOX (15 mg/kg; 3 mg/kg/week), and DOX + AuNPs-treated rats. DOX is known to induce fasting hyperglycemia and hyperinsulinemia in treated rats. Individual treatment of both DOX and AuNPs also promoted liver damage, increased circulatory levels of ALT and AST, and stimulated serum and liver levels of TGs, CHOL, LDL-c, and FFAs. They also stimulated MDA, TNF-α, and IL-6, reduced GSH, SOD, HO-1, and CAT, upregulated mRNA levels of Bax and caspases-3 and -8 and downregulated mRNA levels of Bcl2 in the livers of rats. However, while DOX alone reduced hepatic levels of PPARα, both AuNPs and DOX stimulated mRNA levels of SREBP1, reduced the mRNA, cytoplasmic and nuclear levels of Nrf2, and increased mRNA, cytoplasmic, and nuclear levels of NF-κB. The liver damage and the alterations in all these parameters were significantly more profound when both AuNPs and DOX were administered together. In conclusion, AuNPs exaggerate liver damage, hyperlipidemia, and hepatic steatosis in DOX-treated rats by activating SREBP1 and NF-κB and suppressing the Nrf2/antioxidant axis.

## 1. Introduction

During the last decades, multiple novel chemotherapeutic drugs have been developed with high efficiency to treat hematological and solid tumors and increase the survival of patients [1]. However, the clinical applications of some of these drugs were stopped due to the high chance of systemic and hepatotoxicity [1,2].

Doxorubicin (DOX) is one of the most known anthracycline chemotherapeutic antibiotic drugs that is associated with hepatic damage and steatosis due to its ability to damage DNA and generate high levels of reactive oxygen species (ROS), which subsequently trigger oxidative stress, inflammation, fibrosis, and apoptosis [3,4]. Mechanisms by which DOX promotes hepatic oxidative stress and damage have been recently reviewed in excellent journals and included both enzymatic and non-enzymatic pathways (i.e., activation of NADPH reductase, formation of DOX/Ferrous (Fe^2+^) complex, suppression of the antioxidant transcription factor, the nuclear factor-erythroid factor 2-related factor 2 (Nrf2), depletion of cellular antioxidants (i.e., glutathione (GSH) and other enzymes), and lipid peroxidation [5,6,7,8]. However, the hepatic inflammatory and apoptotic responses associated with DOX are characterized by upregulation/activation of the nuclear factor kappa-beta (NF-κB), higher levels of inflammatory cytokines (i.e., tumor necrosis-α (TNF-α), and interleukine-6 (IL-6), upregulation of Bax/p53, and release of cytochrome-c from the mitochondria [3,4,8,9].

On the other hand, recent advances in fighting tumors have led to the discovery of nanoparticles (NPs) characterized by their unique size and physical and chemical properties. Due to their unique optical properties, small size (1–100 µM), chemical stability, and ease of synthesis, gold nanoparticles (AuNPs) have become the most commonly used NPs used for drug delivery, chemical sensing, and biological imaging [10,11]. However, exposure to AuNPs could occur through inhalation and skin absorption as well as through other routes (i.e., intraperitoneal (i.p.) and intravenous (i.v.)) [10,12]. Humans are at high risk of exposure to AuNPs through dermal contact with jewelry, from dental restorations, and many consuming other products such as toothpaste, lubricants, food packing, cosmetics, automobiles, and beverages [13,14]. However, the safety of these NPs is still uncertain. At the experimental levels, we and others have previously shown that short-term (7 days) exposure to small-sized gold NPs has resulted in a high rate of accumulation in the liver, and this was associated with high levels of ROS, lipid peroxidation, oxidative stress, activation of NF-κB, upregulation of inflammatory cytokines, and liver damage [15,16,17,18,19,20]. However, the effect of long-term exposure to AuNPs, as well as their combined effect with another risk factor on the liver structure and function is not well characterized. These data are alarming and suggest that chronic exposure to these NPs may have several additional adverse effects on the liver, especially with the co-existence of other oxidant drugs. Indeed, AuNPs exaggerated hepatic damage after exposure to lipopolysaccharides (LPS) [21]. In the same manner, it accelerated hepatic steatosis and liver damage in choline-deficient rats [22].

Nonetheless, using AuNPs as a carrier for DOX as an anticancer therapy showed promising results. In this regard, some authors have shown that attaching doxorubicin (DOX)-loaded oligonucleotides (ONTs) to AuNPs (Doxorubicin–Oligomer–AuNP, DOA) was very effective in inhibiting colorectal cancer in vivo and in vitro [23]. However, given the high exposure rate of AuNPs during our daily life, especially among cancer patients, we found it worth evaluating the hepatic toxicity of DOX under the concomitant exposure to small-size AuNPs.

Therefore, compared to their treatments in this study, we demonstrate that AuNPs accelerate DOX-induced hepatic oxidative damage, steatosis, inflammation, and apoptosis after 35 days of treatment. In addition, we are demonstrating that these effects mediate by exaggerating ROS generation, suppressing Nrf2 and antioxidants, activating NF-κB P65, upregulating TNF-α and IL-6, and inhibiting PPARα and FAs oxidation.

## 2. Results

### 2.1. Changes in Food Intake, Body Weights, Glucose and Insulin Levels, and Liver Enzymes

Average weekly food intake was not significantly altered with any treatment (Table 1). Final body weights, fasting plasma glucose, insulin levels, and values of HOMA-IR were not significant, but serum levels of ALT, AST, and γ-GTT were significantly higher in AuNPs-treated rats as compared to control rats (Table 1). Except for final body weights, which were significantly decreased, the levels of all these different endpoints were significantly increased in both DOX and AuNPs + DOX-treated rats compared to control and AuNPs-treated rats (Table 1). Although there were no significant variations in the final body weights, plasma glucose, insulin levels, and levels of HOMA-IR between DOX and DOX + AuNPs-treated rats, the serum levels of ALT, AST, and γ-GTT were significantly higher in DOX + AuNPs-treated rats as compared to DOX-treated rats (Table 1).

### 2.2. Changes in Liver Weights and Serum and Hepatic Lipid Profile

The liver weights and serum levels of TGs, CHOL, LDL-c, and FFAs, as well as serum levels of TGs, CHOL, LDL-c, and FFAs, were significantly increased in AuNPs, DOX, and DOX + AuNP-treated rats as compared to control rats (Table 2). However, the levels of all these parameters were significantly higher in DOX-treated rats as compared to AuNPs-treated rats and in DOX + AuNPs-treated rats as compared to all other groups (Table 2).

### 2.3. Changes in Hepatic Pro-Oxidant, Antioxidant, Inflammatory, and Apoptotic Markers

A significant increase in the hepatic levels of MDA, TNF-α, and IL-6, as well as in the mRNA levels of Bax, caspase-8, and caspase-3 that coincided with a significant reduction in the levels of GSH, SOD, CAT, and HO-1 and mRNA levels of Bcl2, were seen in the livers of AuNPs, DOX, and DOX + AuNPs-treated rats as compared to control rats (Table 3 and Figure 1A–D). The increase in the hepatic levels of MDA, TNF-α, and IL-6 and the mRNA levels of Bax, caspase-8, and caspase-3, as well as the reduction in the mRNA levels of Bcl2 and the levels of all other antioxidant markers, were significantly more profound in both DOX and DOX + AuNPs-treated rats as compared to AuNP-treated rats (Table 3 and Figure 1A–D). However, the maximum significant increase in the levels of MDA, TNF-α, and IL-6 and the transcripts of Bax, caspase-8, and caspase-3 and the maximum significant decrease in the mRNA levels of Bcl2 and levels of GSH, SOD, CAT, and HO-1 were seen in the livers of DOX + AuNPs-treated rats as compared to AuNPs or DOX-treated rats (Table 3 and Figure 1A–D).

### 2.4. Histological Findings

The livers of control rats showed normal liver histology with intact hepatocytes radiating format the central vein. The nuclei of these hepatocytes appeared rounded and normal, and the liver sinusoids looked of normal size (Figure 2A). On the other hand, livers obtained from AuNPs, DOX, and DOX + AuNPs showed various pathologies, including damaged hepatocytes and central vein, the disappearance of sinusoids, increased cytoplasmic cell vacuolization that is filled with the fat droplet, immune cell infiltration, an increased number of necrotic cells having pyknotic karyolysis, and karyorrhexis nuclei (Figure 2B–D). The severity of these abnormalities was higher in DOX as compared to AuNPs-treated rats and was most severe in the livers of DOX + AuNPs-treated rats (Figure 2B–D).

### 2.5. Effect of the Activity and Expression of SREBP1 and PPARα

The transcriptional activity, mRNA levels, and total protein levels of PPARα were not significantly altered in the livers of AuNPS-treated rats but were significantly reduced in the livers of DOX and DOX + AuNPs-treated rats as compared to control rats (Figure 3A,B,E). No significant variations in the transcriptional activity and mRNA levels of PPARα were seen when DOX-treated rats were compared with DOX + AuNPs-treated rats (Figure 3A,B,E). However, protein levels of PPARα were significantly lower in the livers of DOX + AuNPs-treated rats than in DOX-treated rats (Figure 3E). On the other hand, the mRNA levels, transcriptional activity, and total protein levels of SREBP1c were significantly increased in the livers of AuNPs, DOX, and DOX-treated rats as compared to control rats and were significantly higher in the livers of DOX and DOX + AuNPs-treated rats when compared to AuNPs-treated rats (Figure 3C,D,F). The levels of all these SREBP1-related markers were significantly the lowest in the livers of DOX + AuNPs-treated rats compared to DOX or AuNPs-treated rats (Figure 3C,D,F).

### 2.6. Changes in the Transcription, Total Levels, and Nuclear Levels of Nrf2 and NF-κB p65

The mRNA, total and nuclear levels, and nuclear protein content of Nrf2 were significantly decreased, while the mRNA, total and nuclear levels, and nuclear protein content of NF-κB p65 have increased in the livers of AuNPs, DOX, and DOX + AuNPs-treated rats as compared to control rats (Figure 4A–F and Figure 5A,B). However, the most significant decrease in mRNA, as well as the total and nuclear protein levels Nrf2, as well as the maximum reduction in the same parameters of NF-κB p65 were seen in the livers of rats treated with DOX + AuNPs as compared to the individual treatment of DOX and AuNPs (Figure 4A–F and Figure 5A,B).

## 3. Discussion

The finding of this study examined and compared the individual and combined effects of AuNPs and DOX on liver damage in rats. Accordingly, the concomitant administration of both drugs resulted in more severe liver damage in rats associated with hyperlipidemia and hepatic steatosis than in their exposure. In addition, the combined treatment of AuNPs and DOX resulted in an advanced generation of ROS, inflammatory cytokines, and activation/upregulation of NF-κB p6 that coincided with increased suppression of the Nrf2/antioxidant axis. In addition, while DOX suppressed and downregulated SREBP1 and PAPRα, AuNPs only stimulated the transcription and protein levels of SREBP1 alone. Therefore, it can be suggested that AuNPs worsen liver damage and steatosis through their pro-oxidant and pro-inflammatory effects. A graphical abstract summarizing these effects is shown in Figure 6.

ALT, AST, and γ-GTT are the most used markers for assessing liver function and are major indicators of hepatocyte necrosis [24]. A significant increase in circulatory liver markers, as well as other pathological findings such as congestion of the portal and central veins, swelling, degeneration, vacuolization, immune cell infiltration, and necrosis, were reported in the livers of rats individually treated with either DOX or small-sized AuNPs [3,7,15,18]. Similar biochemical and hepatic histological abnormalities were also reported in this study in the DOX or AuNPs-treated animals with a more profound damaging effect associated with DOX treatment. However, the maximum increase in the serum levels of ALT, AST, and γ-GTT, as well as the severest hepatic damage, were observed in the group of rats that were administered the combined treatments of both AuNPs and DOX as compared to their individual effects. Based on these initial data, we become more confident that AuNPs exaggerate the hepatic damage of DOX, probably by its pro-oxidant and pro-inflammatory nature [8].

In addition, DOX treatment significantly reduced rats’ final body weights but significantly increased their weights. On the contrary, treatment with AuNPs had no effects on rats’ bodies and liver weights. In addition, body weights and liver weight were significantly higher in rats co-treated with both drugs. Such increases in the livers weights in both the DOX and DOX + AuNPs-treated rats could be attributed to the increase in the de novo lipid synthesis in the livers of these rats (discussed later) as a result of the synergistic lipogenic effect of both treatments. In support, AuNPs also induced hepatic toxicity and accelerated hepatic lipid synthesis and accumulation in rats fed a choline-free diet [22]. However, since DOX treatment did not change the food intake in rats, it seems reasonable that such a decrease in body weights post-DOX treatment is attributed to the loss of the adipose tissue mass in response to IR and stimulated lipolysis in the adipose tissue. Indeed, DOX reduced rodent body weights by suppressing adipose tissue lipogenesis by inhibiting PPARγ and SREBP1 [25].

Oxidative stress and inflammation are two central-related mechanisms that mediate liver damage in a variety of conditions [26]. Nrf2 is a major antioxidant transcription factor in most cells that prevents oxidative damage through stimulating GSH synthesis as well as the expression of phase-II antioxidant enzymes (e.g., SOD, CAT, and HO-1) [27]. Opposing this, NF-κB is the major pro-oxidant and inflammatory transcription factor that stimulates inflammation by upregulating numerous cytokines and adhesive molecules (e.g., IL-1β, TNF-α, and IL-6), which in turn promote macrophage infiltration, inflammation, and generation of ROS [28]. ROS and NF-κB can stimulate each other, leading to a vicious activation cycle [28]. In addition, Nrf2 and NF-κB are negatively cross-talked with each other, where each of them can suppress the activity of the other [7,29]. In the liver, the metabolism of DOX upregulates numerous ROS-generating enzymes that promote oxidative and inflammatory damage [7]. Short and long-term treatment with DOX was associated with higher levels of hepatic ROS, TNFα, and IL-6, over-activation of NF-κB, and reduced activities and expression of Nrf2 [3,4,6,8,30,31,32]. In addition, short-term treatment with AuNPs promotes hepatic and renal oxidative stress and inflammation by generating ROS and inflammatory cytokines, scavenging GSH, consuming SOD and CAT, and activating NF-κB [15,16,18,19,21]. Of note, the effect of AuNPs on hepatic and systemic Nrf2 signaling is not yet established in humans or animals.

In the current study, oxidative stress and inflammation were also evidenced in the livers of both groups of rats, which were administered AuNPs or DOX either individually or in combination. This picture included higher levels of ROS, MDA (a marker of lipid peroxidation), TNF-α, IL-6, mRNA, and nuclear activities of NF-κB p65. In addition, the livers of both treated groups also showed a significant reduction in the hepatic levels of GSH, SOD, HO-1, and CAT and reduced transcription and the nuclear activities of Nrf2. However, the changes in the levels of all these biochemical endpoints were more significant and profound in the groups of rats which received the combination treatment, which suggests that a combination of AuNPs and DOX synergistically stimulates liver damage by exaggerating the oxidative stress and inflammatory responses. Yet, these data cannot precisely identify whether Nrf2 locates upstream or downstream of NF-κB during this toxicity. However, we have recently shown that DOX could promote liver damage by modulating these pathways through suppressing SIRT1, which normally activates Nrf2/antioxidant axis and suppresses NF-κB p65 and inflammation [8]. Hence, targeting SIRT1 could represent a novel target to reveal the hepatic toxicity induced by AuNPs.

Apoptosis is the cell death process triggered by oxidative and inflammatory damage to the cells [33]. ROS, NF-κB p65, and TNF-α can activate both the extrinsic and intrinsic cell apoptosis pathways [34,35,36,37,38]. Cell apoptosis can be intrinsic or extrinsic and is imitated by activating several cytoplasmic endoproteases called caspases (i.e., caspases-3, 8, and 9) [39]. Extrinsic cell death is initiated by binding a legend such as TNF-α and Fas with a death surface receptor (TNFR and FasR), which stimulates the activation of caspase-8/3 [40]. On the other hand, intrinsic cell death is induced by an imbalance in the apoptotic (i.e., Bax)/ant apoptotic (Bcl2) proteins, which allow damaging of the mitochondria membranes and the release of cytochrome-c, which in turn stimulates capaspses-9/3 [33,40]. In addition, DNA damage, ROS, and NF-κB are potent stimulators of intrinsic cell death through upregulating p53, which subsequently increases the expression of Bax [41]. In this study, and associated with the high levels of ROS and levels of TNF-α and higher activation of NF-κB p65, intrinsic cell apoptosis was also activated in the livers of both DOX and AuNPs-treated rats supported by the higher mRNA levels of Bax, caspases-8 and-3 and reduced expression of Bcl2 in the livers of both groups. This is in line with many previous studies showing similar apoptotic effects of DOX in rodents’ livers [4,7,8]. In addition, Kassab et al. [42] have previously shown apoptotic effects of naked AuNOs (10–15 nm) in the liver of rats mediated by the upregulation of p53, downregulation of Bcl2, and inducing TNF-α. However, a more significant reduction in the levels of Bcl2 and a more significant increase in the expression of Bax and all measured caspases were observed in the livers of rats co-treated with AuNPs and DOX, suggesting a cooperative apoptotic effect. This could be explained by the higher levels of ROS and TNF-α as well as the increased transcription and activation of NF-κB p65 generated in the livers of these rats under this combination treatment. Supporting this, co-treatment with AuNPs enhanced and facilitated apoptotic hepatocyte cell death in LPS-treated rats [21].

We have also observed that rats co-administered with the combined treatment of both AuNPs and DOX developed hyperlipidemia, and their liver showed an advanced stage of hepatic steatosis compared to the individual treatment of each drug. On the one hand, associated with the previously discussed increase in total body weight, we have noticed that DOX-treated rats had a T2DM phenotype that is characterized by fasting hyperglycemia, hyperinsulinemia, high HOMA-IR, and hyperlipidemia. In addition, they developed hepatic steatosis and had increased levels of FFAs, TGs, and CHOL. As confirmed by others, such an increase in FFAs hepatic content could be explained by the peripheral IR and increased adipose tissue lipolysis [25]. Indeed, DOX is a risk factor for the development of T2DM and NAFLD [43,44]. Short and long-term DOX treatment can induce IR, hyperglycemia, and hyperlipidemia, increase hepatic FFAs contents, stimulate hepatic de novo lipogenesis, and promote NAFLD through several pathways, including stimulating adipose tissue lipolysis by activating PPARα [25]. Suppressing adipose tissue and muscular PPARγ and AMPK expression and activation reduces adipogenesis, insulin sensitivity, and glucose disposal [25,45]. Furthermore, DOX reduces the release of FFAs and stimulates TGs synthesis by suppressing the peroxisomal β-oxidation and inhibiting hepatic adipose triglyceride lipase (ATGL), which normally stimulates mitochondria β-oxidation [22,45].

However, hyperlipidemia with increased hepatic levels of CHOL and TGs and fat vacuoles were seen in AuNPs-treated rats, suggesting a hyperlipidemic effect independent of modulating peripheral glucose or insulin signaling. Nonetheless, it seems reasonable that AuNPs aggravated the ballooning of the hepatocytes by increasing oxidative stress and inflammation, both of which facilitate the progression from simple steatosis to non-alcoholic hepatitis. Therefore, these data could be of much interest as they provide a clear warning for the steatotic effect of AuNPs, especially with the existence of other factors that stimulates hepatic lipid accumulation. Supporting this, treatment with AuNPs accelerated the development of NAFLD and NASH in HFD-fed animals [22].

However, we were interested in revealing the precise hepatic molecular mechanisms responsible for the steatosis effect of both AuNPs and DOX. For this reason, we have targeted PPARα and PPARγ, as well as SREBP1c, which are major regulators involved in hepatic lipid metabolism [41,46]. In general, SREBP1c is the major transcription factor responsible for FAs and TGs synthesis by regulating a bunch of lipogenic genes such as fatty acid synthase (FAS) and cetyl-CoA carboxylase 1 (ACC-s) [46]. On the other hand, PPARα reduces TGs accumulation by stimulating mitochondria β oxidation by regulating the L-carnitine system [47,48,49,50,51]. The treatment with PPARα agonists is an effective therapy to alleviate liver damage and NAFLD [48,49,50,51]. As discussed above, DOX treatment stimulated adipose tissue PPARα [25]. Yet the effect of DOX and AuNPs on the expression of hepatic SREBP1 and PPARα is still unknown.

DOX significantly reduced the expression of PPARα in cultured podocytes and cardiomyocytes of intoxicated mice [41,52,53]. In the same line, we have also found a significant reduction in transcriptional activity, mRNA, and protein levels of PPARα in the livers of DOX-treated rats of this study that was concomitantly associated with increased transcription of SREBP1c. These data indicate that DOX has a stimulatory role in lipid synthesis by stimulating de novo lipogenesis and suppressing FA oxidation through stimulating SREBP1 and downregulating/inhibiting PPARα. Interestingly, the hepatic PPARα is a potent antioxidant and anti-inflammatory transcription factor that prevents liver damage by stimulating the Nrf2/antioxidant axis and suppressing NF-κB and inflammatory cytokine production [47,50,54]. Such a reduction in the activities of PPARα via DOX treatment might also explain why the livers of these DOX-treated rats showed reduced expression and activation of Nrf2 and, in parallel, reduced those of NF-κB, as discussed before. Herein, we are showing evidence that the steatotic, pro-oxidant, and pro-inflammatory role of DOX in the liver of rats is mediated at least by suppressing PPARα, which represents a novel therapeutic target.

In addition, our data indicate that AuNPs stimulate de novo lipid synthesis only through stimulating the expression of SREBP1c with no obvious effect on the expression/activity of PPARα. Indeed, treatment with AuNPs showed no effect on the transcriptional activity, mRNA, and protein levels of PPARα. Interestingly, we have observed a significant reduction in protein levels of PAPRα in the livers of rats that concomitantly administered AuNPs and DOX. Such a reduction could be explained by the higher levels of ROS generated by the combined treatment compared to AuNPs alone. Therefore, these data suggest that AuNPs alone stimulate lipogenesis and stimulate ROS generation, whereas they may also repress PPARα if co-treated with DOX-treated rats, which by itself can activate both damaging pathways. However, the combinations of all these lipid-related mechanisms contributed significantly to the obvious increase in the levels of hepatic FFAs, TGs, and CHOL in the livers of rats who received the combined treatment and may explain the severe accumulation of fat droplets and hepatocyte ballooning appeared in the livers of these rats. In general, hyperglycemia, ROS, and TNF-α are portent activators of SREBP1 [53,55,56,57]. Therefore, all these factors could have contributed significantly to the observed stimulatory effects of both AuNPs and DOX on the hepatic expression of SREBP1 in the treated rats. However, the precise mechanism by which DOX inhibits PPARα remains unknown and cannot be concluded from this study.

Although we are showing interesting data, our data still have some limitations. Importantly, this study examined the effect of AuNPs on DOX-induced toxicity for a limited period (i.e., 35 days after treatment). Therefore, it will be more valuable to examine the combined effects of these drugs at different time intervals post-administration. In addition, as demonstrated in this study, the effects of DOX and AuNPs seem to involve numerous signaling pathways. Hence, further studies using animals or cells deficient with some key regulators (i.e., PPARα, Nrf2, and NF-κB) will widen our knowledge about the exact molecular mechanisms regulated by AuNPs and DOX.

## 4. Materials and Methods

### 4.1. Animals

Adult Wistar male rats (210 ± 20 g) aged 10 weeks old were supplied from the Experimental Animal Care Center at King Saud University, Riyadh, KSA. All animals were always housed in plastic cages in a controlled room (temperature = 22 ± 5 °C, humidity= 55 ± 5%, and a 12 h light/dark cycle and had free access to diet and water ad libitum. Experiments were approved by Research Ethics Committee at King Saud University (Ethics Reference No: KSU-SE-21-11), Riyadh, Saudi Arabia, and followed the Animal Research Reporting of in vivo Experiments (ARRIVE) guidelines.

### 4.2. Drugs

AuNPs (10 nm) were (Cat. No. MKN-Au-010) purchased from IPEX Corp, MKNano, Toronto, Canada, and were identified by electron microscopy. DOX hydrochloride (Cat. NO. D1515) was purchased from Sigma Aldrich, St. Louis, MO, USA.

### 4.3. Experimental Design

A total of 32 rats were included in this study and were divided into 4 groups (each of 8 rats) as follows: (1) control rats: daily treated with 250 µL of 0.9% normal saline as a vehicle for 5 weeks; (2) AuNPs-treated rats: daily treated with 250 µL of 10 nm AuNPs for 5 weeks; (3) DOX-treated rats: treated with an accumulative dose of DOX solution (15 mg/kg; 3 mg/kg/week) and co-treated with 250 µL of 0.09% normal saline (4) DOX + AuNPs-treated rats: treated with DOX (15 mg/kg; 3 mg/week) and co-treated daily with 250 µL of 10 nm AuNPs (i.p.) for 5 weeks. All treatments were administered intraperitoneal (i.p.). The dose and route of treatment with DOX were selected based on the previous study, which showed impaired glucose tolerance, IR, and hyperglycemia in rats [45]. It also induced hepatic oxidative damage, suppressed Nrf2, and upregulated NF-κB in the livers of rats [3,8,58]. On the other hand, the dose of AuNPs was selected based on previous evidence of hepatic damage by stimulating the generation of ROS, upregulation of inflammatory cytokines, and activation of NF-κB [15,18,59].

### 4.4. Tissue and Blood Collection

On day 36, all rats were anesthetized using a single dose of ketamine/xylazine hydrochloride mixture (80/10 mg/kg, *v*:*v*), and their blood samples were directly collected by cardiac puncture into plain tubes and EDAT-containing tubes and then centrifuged at 1300× *g* (room temperature 10 min) to collect serum and plasma, respectively. These samples were always stored at −20 °C until use. Next, all rats were sacrificed by cervical dislocation, and their livers were rapidly isolated on ice. The livers were washed with an ice-cold phosphate-buffered saline (PBS) (pH = 7.4) and cut into smaller pieces, which were each 3–4 mm. Parts of these livers were fixed in specific fixatives for the histological or electron microscopy studies. All other parts were kept at −80 °C and used later for the rest of the experiments.

### 4.5. Extraction of Hepatic Lipids from the Freshly Collected Livers

Parts of the newly organized livers (*n* = 8/group) were directly used to extract lipids, adopting the methanol:chloroform: normal saline method described by Folch et al. [60]. Briefly, parts of the liver weighing 0.25 g were homogenized in 10 mL of a methanol: chloroform solution (1:2, *v*/*v*) for 1 h at 4 °C. The mixture was filtrated, and 2 mL of normal saline was added. The mixture was vortexed, which was followed by centrifugation (1200× *g*; 10 min). The lower organic layer containing the dissolved lipids was isolated, and the solvent was evaporated. The collected lipids were dissolved in 0.5 mL of isopropanol and used for different lipid quantifications.

### 4.6. Preparation of Liver Homogenates and Nuclear Fractionation

Liver parts from each rat (70 mg) were homogenized in 0.5 mL ice-cold PBS (pH = 7.4) and then centrifuged at 12,000× *g* for 15 min. The supernatants were isolated and distributed into new Eppendorf tubes. At the same time, the cytoplasmic/nuclear fractions were prepared using a commercial kit (NE-PER Cat. No. 78833, ThermoFisher, Waltham, MA, USA). All isolated supernatants and cellular fractions were kept at −80 °C until use.

### 4.7. Biochemical Analysis in the Plasma

The glucose levels in the plasma were measured using an assay kit (Cat No. Ab65333, Abcam, Cambridge, UK). Plasma insulin levels were assayed by ELISA (Ca., No. MBS045315). The homeostasis model of insulin resistance (HOMA-IR) was manually calculated from the following equation (fasting insulin (µU/L) × fasting glucose (nmol/L)/405) [61]. All measurements were conducted as per the provided instructions.

### 4.8. Biochemical Analysis of the Serum and Tissue Homogenates

Serum levels of alanine aminotransferase (ALT), gamma-glutamyl transferase (γ-GTT), and aspartate aminotransferase (AST) were measured using assay kits (Cat. No. MBS269614, MyBioSource, San Diego, CA, USA; Cat. No. MBS9343646, MyBioSource, San Diego, CA, USA; and Cat. No. CSB-E13023r-1, Cosmo Bio, Carlsbad, CA, USA), respectively. Homogenate levels of malondialdehyde (MDA), interleukine-6 (IL-6), total reduced glutathione (GSH), superoxide dismutase (SOD), tumor necrosis factor-alpha, heme oxygenase-1 (HO-1), and catalase (CAT) were measured using assay kits purchased from MyBioSource, San Diego, CA, USA (Cat. No. MBS2540407; Cat. No. MBS269892, Cat. No. MBS265966, Cat. No. MBS036924, and Cat. No. MBS2507393, Cat. No. MBS764989; and Cat. No. MBS006963, MyBioSource, San Diego, CA, USA, respectively). The transcriptional activities of PPARα and SREBP1 in the nuclear fractions were assayed using an assay kit (Cat. No. Ab133107 and Cat. No. ab133125, Abcam, Cambridge, UK, respectively). The concentrations of NF-κB p65 and Nrf2 in the cytoplasmic and nuclear fractions were determined using ELISA kits (Cat. No. MBS2505513 and Cat. No. MBS752046, MyBioSource, San Diego, CA, USA, respectively). All procedures were conducted per each kit’s instructions for *n* = 8 samples/group and per the manufacturer’s recommendations and instructions.

### 4.9. Real-Time PCR (qPCR)

Real-time PCR was conducted to measure the mRNA transcript levels of SRBEP1, PAPRα, Nrf2, NF-κB, Bcl2, Bax, caspase-3, and β-actin (a reference gene). Primers have been previously described by us and others [8,62]. All primers were purchased and provided by ThermoFisher and were previously used by us and others. For this part, RNA isolation was conducted using the TRIZOl reagent, and the purity of the isolated RNA was measured at the absorbance of 260/280 using a nanodrop spectrophotometer. A commercial kit synthesized the first-strand cDNA (Cat. No. GE27-9261-01, Roche Diagnostic Company, Indianapolis, IN, USA). Q-PCR was performed using the Sofas Evergreen master mix kit (# 172–5200, Biorad, Hercules, CA, USA) on a CFX69 real-time PCR machine (Biorad) following the steps mentioned in the kit. In brief, the reaction mixture/well (20 µL) contained the following ingredients: 2 μL cDNA (50 ng/well); 10 µL of the master mix reagent; 0.2 µL of the forward primer (500 nM/each); 0.2 µL of the reverse primer (500 nM/each); and 7.6 µL nuclease-free water. Amplification steps were heating (1 cycle/98 °C/30 s), denaturation (40 cycles/98 °C/5 s), annealing (40 cycles/60 °C/5 s), and melting (1 cycle/5 s/80–95 °C). The relative expression of all targets was normalized to the expression of the reference gene, β-actin.

### 4.10. Western Blotting

The Western blotting protocol was described in detail in our previous studies [8]. Briefly, total cytoplasmic and nuclear proteins were prepared in the loading dye buffer to a final concentration of 2 µg/µL. All tubes were then boiled at 100 °C for 5 min. Equal protein concentrations (60 µg/well) were separated by SDS-PAGE, transferred to a nitrocellulose membrane, and then incubated with the primary antibodies against SREBP1 (Cat. No. sc-13551, 125 kDa, Santa Cruz Biotechnology, Dallas, TX, USA), PPARα (Cat. No. sc-398394, 55 kDa, Cell Signaling Technology, Danvers, MA, USA), Nrf2 (Cat. No. 12721, 100 kDa, Cell Signaling Technology, Danvers, MA, USA), NF-κB (Cat. No. sc-8008, Santa Cruz Biotechnology, Dallas, TX, USA), β-actin (Product No. 4970, 45 kDa, Cell Signaling Technology, Danvers, MA, USA), and Lamine A (nuclear loading control) (Product No. 86846, 74 kDa, Cell Signaling Technology, Danvers, MA, USA). The membranes were then washed with the washing buffer and incubated with the HRP peroxidase-conjugated 2nd antibody. The developed bands were scanned, visualized, and photographed using the C-Di Git blot scanner (LI-COR, Lincoln, NE, USA) and its provided software after incubating each membrane with the chemiluminescence west-pico reagent (Cat. No. 34580, Thermo Fisher, Piscataway, NJ, USA).

### 4.11. Histological Evaluation

This has been conducted as reported in our previous report [4]. For the histology part, formalin-preserved livers were deparaffinized in xylene with decreasing levels of ethanol (i.e., 100%, 90%, and 70%). The liver tissues were embedded in wax and then cut using a rotatory microtome at a thickness of 4–5 µM. After this, all tissues were stained with Harris hematoxylin/glacial acetic acid solution and then de-stained with a 1:400 *v*/*v* HCL/ethanol (70%) mixture. A single drop of eosin was then added to each section. All tissues were then covered with a mounting media and a coverslip. All sections were photographed using a light microscope at 200×.

### 4.12. Statistical Analysis

GraphPad Prism analysis software (Version 8, San Diego, CA, USA) was used for the statistical analysis of all data. A Kolmogorov–Smirnov test was utilized to test the normality. Analysis was performed using the 1-way ANOVA test. The levels of significance were determined using Tukey’s test as post hoc (*p* < 0.05). All data were expressed in the results as means ± standard deviation (SD).

## 5. Conclusions

Our data warn about the adverse hepatic toxic and steatosis effects of contamination with AuNPs during the DOX therapy in rats. Based on our data, it seems that AuNPs and DOX work synergistically to promote hepatic damage and steatosis by upregulating numerous lipid-related lipogenic genes and inflammatory mediators as well as suppressing FAs oxidation and antioxidant genes. However, these data encourage further studies at the clinical level.

## Figures and Tables

**Figure 1 molecules-28-00796-f001:**
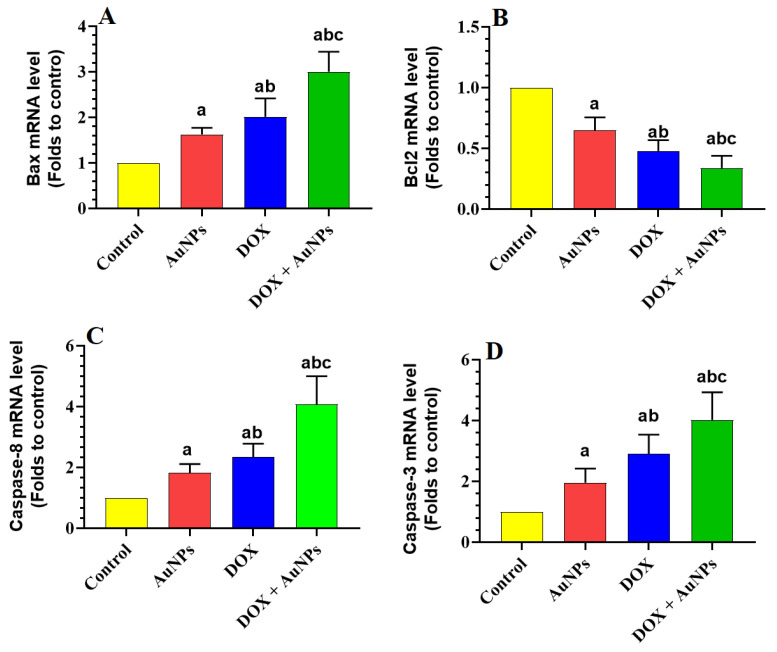
Effects of the individual or combined treatments with gold nanoparticles (AuNPs) and/or doxorubicin (DOX) on the transcription of some selected apoptotic and anti-apoptotic markers in the liver of all experimental groups. (**A**): Bax mRNA, (**B**): Bcl2 mRNA, (**C**): Caspase-8, and (**D**): Caspase-3. Data were analyzed by 1-way ANOVA followed by Tukey’s test. Values are presented as means ± SD (*n* = 8/group). Significance was considered at *p* < 0.5. ^a^: vs. control rats; ^b^: vs. AuNPs-treated rats; ^c^: vs. DOX-treated rats.

**Figure 2 molecules-28-00796-f002:**
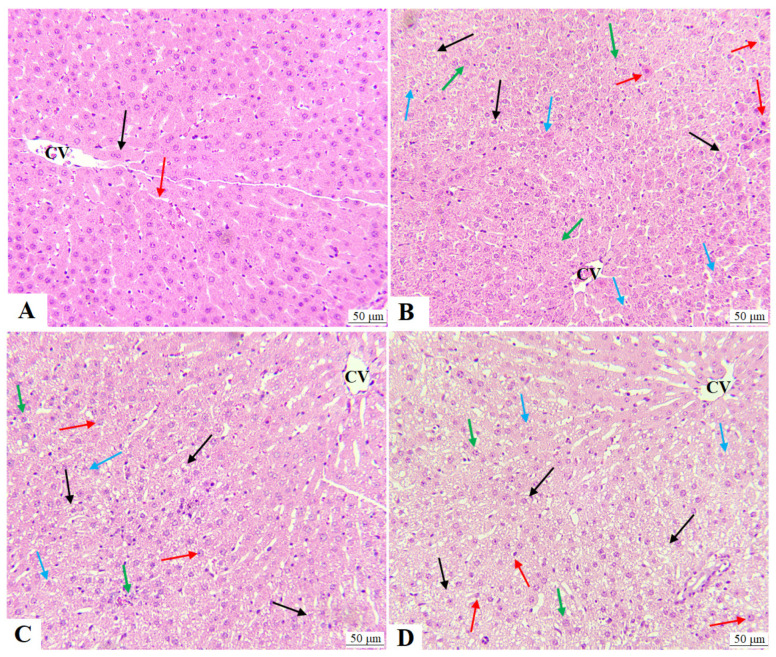
Histological pictures were obtained for the livers from all experimental groups and stained with hematoxylin and eosin (H&E). All photos were captured at 200×. (**A**) is a control rat showing a normal central vein (CV) in which normally appeared hepatocytes with round nuclei (black arrow). In addition, the size of the sinusoid appeared normal (red arrow). (**B**–**D**) represent AuNPs, DOX, and DOX + AuNPs-treated rats, respectively, and show a damaged CV and increased number of hepatocytes with vacuolated and lipid-filled cytoplasm (black arrow). In addition, the nuclei of the hepatocytes of all these groups of rats showed pyknosis (red arrow), karyolysis (blue arrow), and karyorrhexis (green arrow). Note the increase in all these abnormalities in DOX + AuNPs-treated rats as compared to other groups.

**Figure 3 molecules-28-00796-f003:**
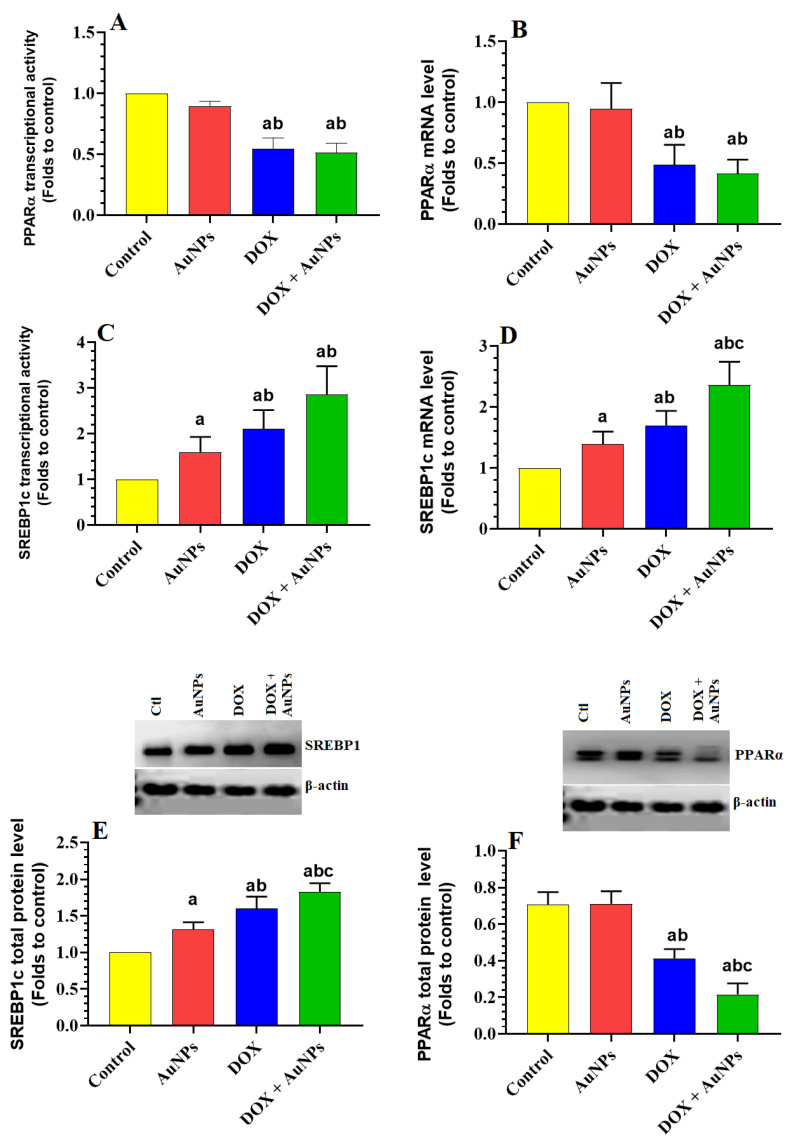
Effects of the individual or combined treatments with gold nanoparticles (AuNPs) and/or doxorubicin (DOX) on the transcription, total levels, and nuclear transcriptional activity of PPARα (**A**,**B**,**F**) and SREBP1 (**C**,**D**,**E**) in the liver of all experimental groups. Data were analyzed by 1-way ANOVA followed by Tukey’s test. Values are presented as means ± SD (*n* = 8/group). Significance was considered at *p* < 0.5. ^a^: vs. control rats; ^b^: vs. AuNPs-treated rats; ^c^: vs. DOX-treated rats.

**Figure 4 molecules-28-00796-f004:**
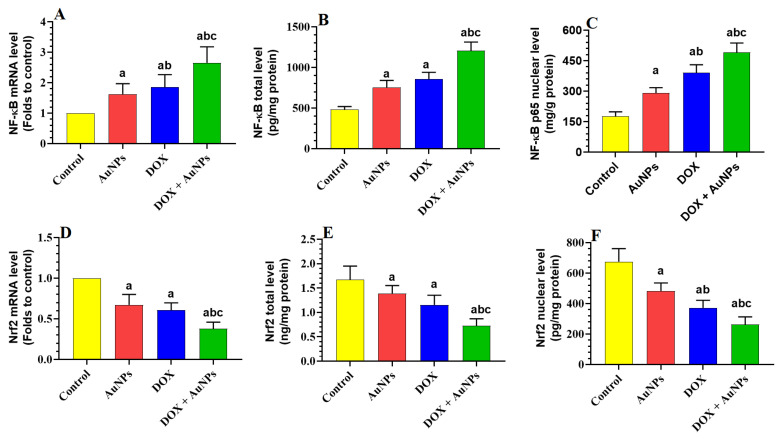
Effects of the individual or combined treatments with gold nanoparticles (AuNPs) and/or doxorubicin (DOX) on the transcription, total levels, and nuclear levels of NF-kB (**A**–**C**) and Nrf2 (**D**–**F**) in the liver of all experimental groups. Data were analyzed by 1-way ANOVA followed by Tukey’s test. Values are presented as means ± SD (*n* = 8/group). Significance was considered at *p* < 0.5. ^a^: vs. control rats; ^b^: vs. AuNPs-treated rats; ^c^: vs. DOX-treated rats.

**Figure 5 molecules-28-00796-f005:**
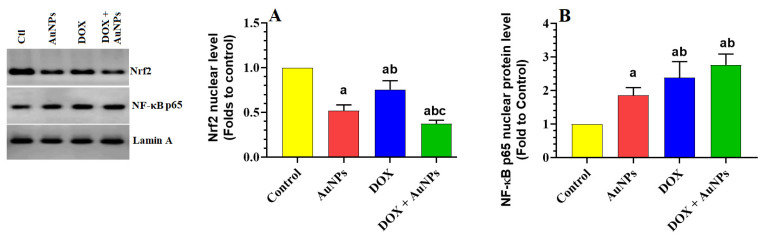
Effects of the individual or combined treatments with gold nanoparticles (AuNPs) and/or doxorubicin (DOX) on the nuclear levels of Nrf2 (**A**) and NF-κB p65 (**B**) in the liver of all experimental groups. Data were analyzed by 1-way ANOVA followed by Tukey’s test. Values are presented as means ± SD (*n* = 8/group). Significance was considered at *p* < 0.5. ^a^: vs. control rats; ^b^: vs. AuNPs-treated rats; ^c^: vs. DOX-treated rats.

**Figure 6 molecules-28-00796-f006:**
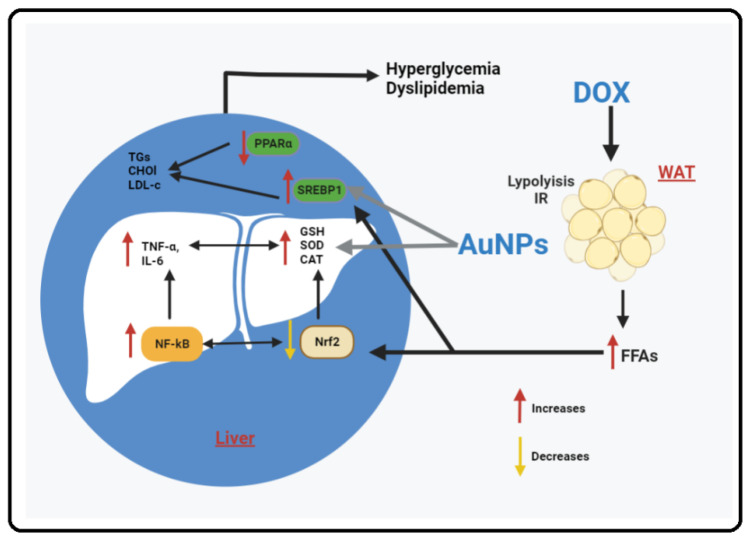
A graphical abstract demonstrating the synergistic hepatic toxic effect of doxorubicin (DOX) and small-sized gold nanoparticles (AuNPs) in rats. In the figure, DOX triggers insulin resistance (IR) and lipolysis and the white adipose tissue (WAT), which leads to an increase in the influx of free fatty acid (FFAs) to the livers and induces oxidative stress and inflammation by the generation of reactive oxygen species (ROS) and scavenging glutathione (GSH), superoxide dismutase (SOD), and catalase (CAT), suppression of Nrf2, activation of NF-κB p65, and increasing levels of inflammatory mediators such as tumor necrosis factor-α (TNF-α) and interleukin-6 (IL-6). In addition, DOX (possibly through increasing hepatic FFAs) can directly stimulate SREBP1 and inhibit PPRAα. On the other hand, AuNPs affect liver health and lipogenesis by acting on oxidative stress and inflammatory markers (similar to the effect of DOX) as well as inhibiting SREBP1.

**Table 1 molecules-28-00796-t001:** Effects of the individual or combined treatments with gold nanoparticles (AuNPs) and/or doxorubicin (DOX) on body weights, food intake, liver enzymes, and plasma glucose and insulin levels in all groups.

Parameter	Control	AuNPs	DOX	DOX + AuNPs
Final body weight (g)	265 ± 24	259 ± 28	221 ± 15 ab	226 ± 18 ab
Average weekly food intake (g/group)	1673 ± 123	1537 ± 143	1703 ± 154	1607 ± 162
Serum				
γ-GTT (U/L)	28.4 ± 3.7	46.7 ± 5.3 a	76.4 ± 5.9 ab	98.3 ± 6.9 abc
ALT (U/L)	39.8 ± 5.4	65.6 ± 6.5 a	88.5 ± 7.1 ab	123.5 ± 11.4 abc
AST (U/L)	47.2 ± 5.1	71.3 ± 6.3 a	95.1 ± 8.4 ab	131.3 ± 10.4 abc
Plasma				
Fasting glucose (nmol/L)	4.6 ± 0.4	4.4 ± 0.6	6.4 ± 0.5 ab	6.6 ± 0.9 ab
Fasting insulin (µIU/mL)	6.0 ± 1.4	6.2 ± 1.6	10.9 ± 2.4 ab	11.3 ± 2.9 ab
HOMA-IR	1.3 ± 0.39	1.18 ± 3.2	3.1 ± 0.41 ab	3.3 ± 0.9 ab
Final body weight (g)	265 ± 24	259 ± 28	221 ± 15 ab	226 ± 18 ab

Data were analyzed by 1-way ANOVA followed by Tukey’s test. Values are presented as means ± SD (*n* = 8/group). Significance was considered at *p* < 0.5. a: vs. control rats; b: vs. AuNPs-treated rats; c: vs. DOX-treated rats. HOMA-IR: the homeostasis model of insulin resistance = fasting insulin (µU/L) × fasting glucose (nmol/L)/405.

**Table 2 molecules-28-00796-t002:** Effects of the individual or combined treatments with gold nanoparticles (AuNPs) and/or doxorubicin (DOX) on liver weights and serum/hepatic lipids in all groups.

Parameter		Control	AuNPs	DOX	AuNPs + DOX
	Liver weight	12.1 ± 0.66	13.9 ± 0.56 ^a^	15.6 ± 0.79 ^ab^	16.9 ± 0.9 ^abc^
**Serum**	TGs (mg/dL)	33.1 ± 3.4	47.8 ± 4.8 ^a^	59.1 ± 4.2 ^ab^	72.3 ± 5.8 ^abc^
	CHOL (mg/dL)	76.8 ± 5.3	89.5 ± 5.4 ^a^	98.3 ± 7.3 ^ab^	121 ± 9.2 ^abc^
	LDL-c (mg/dL)	34.3 ± 3.9	45.4 ± 4.3 ^a^	58.9 ± 4.7 ^ab^	69.3 ± 6.1 ^abc^
	FFAs (µmol/L)	422 ± 39.4	389 ± 41.2	745 ± 57.8 ^ab^	719 ± 61.3 ^ab^
**Liver**	Triglycerides (µg/g)	3432 ± 288	4552 ± 382 ^a^	5434 ± 392 ^ab^	6729 ± 428 ^abc^
	CHOL (µg/g)	4902 ± 309	6022 ± 519 ^a^	7022 ± 538 ^ab^	8192 ± 632 ^abc^
	FFA (µmol/g)	112.4 ± 9.5	122 ± 13.5	258 ± 17.8 ^ab^	263± 21.9 ^ab^

Data were analyzed by 1-way ANOVA followed by Tukey’s test. Values are presented as means ± SD (*n* = 8/group). Significance was considered at *p* < 0.5. ^a^: vs. control rats; ^b^: vs. AuNPs-treated rats; ^c^: vs. DOX-treated rats.

**Table 3 molecules-28-00796-t003:** Effects of the individual or combined treatments with gold nanoparticles (AuNPs) and/or doxorubicin (DOX) on selected markers of oxidative stress and inflammation in the liver of all experimental groups.

Parameter	Control	AuNPs	DOX	DOX + AuNPs
MDA (µM/mg tissue)	1.05 ± 0.14	1.88 ± 0.22 ^a^	2.13 ± 0.28 ^ab^	2.83 ± 0.41 ^abc^
GSH (µg/mg tissue)	29.5 ± 2.1	20.4 ± 1.9 ^a^	16.8 ± 1.3 ^ab^	11.9 ± 0.92 ^abc^
SOD (U/mg tissue)	8.9 ± 0.92	6.5 ± 0.54 ^a^	5.0 ± 0.49 ^ab^	3.9 ± 0.21 ^abc^
CAT (U/mg tissue)	3.65 ± 0.22	2.2 ± 0.41 ^a^	1.63 ± 0.17 ^ab^	1.1 ± 0.15 ^abc^
HO-1 (ng/mg tissue)	7.4 ± 0.81	5.1 ± 0.49 ^a^	4.2 ± 0.68 ^ab^	3.4 ± 0.53 ^abc^
TNF-α (pg/mg tissue)	13.5 ± 2.1	24.7 ±3.1 ^a^	33.4 ± 3.3 ^ab^	45.3 ± 4.6 ^abc^
IL-6 (pg/mg tissue)	41.4 ± 4.9	63.2 ± 5.8	73.2 ± 4.9 ^ab^	89.2 ± 3.2 ^abc^

Data were analyzed by 1-way ANOVA followed by Tukey’s test. Values are presented as means ± SD (*n* = 8/group). Significance was considered at *p* < 0.5. ^a^: vs. control rats; ^b^: vs. AuNPs-treated rats; ^c^: vs. DOX-treated rats.

## Data Availability

The datasets used and analyzed during the current study are available from the corresponding author upon reasonable request.

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
