# Peer review of "Concomitant Sub-Chronic Administration of Small-Size Gold Nanoparticles Aggravates Doxorubicin-Induced Liver Oxidative and Inflammatory Damage, Hyperlipidemia, and Hepatic Steatosis"

_molecules, 2023, doi:10.3390/molecules28020796_

Round 1
Reviewer 1 Report
The present study proposed by Alshammari and colleagues reported the the adverse hepatic toxic and steatosis effects of concomitant administration of AuNPs during the DOX-therapy in rats. The study is scientifically interesting and quite meaningful. They showed AuNPs aggravated DOX-stimulated hepatic damage and steatosis by up-regulating kinds of lipid-related genes and inflammatory mediators, and by inhibiting FAs oxidation and antioxidant genes. Generally speaking, the study is well conceived and clearly described, the techniques and statistical analysis were appropriated, and the conclusions were supported by the results. However, there are minor points to address:
Major:
1. Most of the study were descriptive, and little mechanistic insights were revealed in the present study. The authors displayed AuNPs aggravated DOX-stimulated adverse effects via Nrf2 and NF-κB by PCR and WB to analyze the mRNA and protein levels. However, this is not the direct evidence. I suggest the authors to assess the Nrf2 and NF-κB pathways by phosphorylation or dephosphorylation via WB.
Minor:
1. Line 86, “Adult male Wistar male rats ” should be “Adult Wistar male rats”, “Adult male Wistar rats”, or “Male adult Wistar rats”.
2. Line 94, “purchased from (IPEX Corp, Canada)” should be “purchased from IPEX Corp (Canada)” .
3. In 2.3 section, control rats: daily treated with 250 µl of 0.9% normal saline ; 2) AuNPs-treated rats: daily treated with 50 L of 10 nm AuNPs ; 3) DOX-treated rats: treated with an accumulative dose of DOX solution (15 mg/kg) and co-treated with 250 µl of 0.09% normal saline 4) DOX + AuNPs-treated rats: treated with DOX (15 mg/kg) and co-treated daily with 250 µl of 10 nm AuNPs. In a standard design, the volume of each group should be the same, however in the present study the authors did not follow this principle.
4. In 2.4 section, I don’t think it is necessary to discuss the “Dose selection” in this part. I suggest the authors combine section 2.3 and 2.4, or discuss “Dose selection” in the discussion section.
5. Line 119, “All rats were authenticated by cervical dislocation” . If I have understood correctly, “authenticated” should be “sacrificed”.
6. In section 2.5, the authors described how to collect tissue and serum samples which would be used later. However, they missed to introduce how to get plasma.
7. Line 166, “Also, were all provided by ...” This sentence did not have subject.
8. Line 181, “(Al-Qahtani et al., 2022)” should be deleted and “briefly” should be “Briefly”.
9. Line 1987, “(Alsham- 197 mari et al., 2021)” should be deleted.
10. Panels of each figure should be correctly numbered, eg: Figures 2 and 4.
11. Lines 365-366, the sentence is incomplete.
Author Response
Reviewer 1
Comment: Most of the study were descriptive, and little mechanistic insights were revealed in the present study. The authors displayed AuNPs aggravated DOX-stimulated adverse effects via Nrf2 and NF-κB by PCR and WB to analyze the mRNA and protein levels. However, this is not the direct evidence. I suggest the authors to assess the Nrf2 and NF-κB pathways by phosphorylation or dephosphorylation via WB.
Response: Dear reviewers, Thank you very much for this comment. This is very valuable. However, in this study, we just provided evidence that AuNPs act by exaggerating hepatic steatosis and hyperlipidemia in the livers of DOX-treated rats the oxidative and inflammatory response, and through modulating Nrf2 and NF-kb. This highlight the regulatory role of these AUNPs on these pathways, and this was supported by the qPCR, western blot, and biochemical measurements of these factors. However, as the reviewer knows, the regulation of these factors has many upstream mechanisms, such as AMPK, which can phosphorylate, and SIRT1, which can deacetylate these factors, etc. In addition, some other mechanisms can regulate these factors. So, to figure out the upstream mechanism, we need to study the expression of AMPK and SIRT1, as well as other regulatory proteins, as well as phosphorylation and deacetylation rates, both factors. This will be a lot of work which requires months of work and need a special budget, which we don’t have at this time. We need to continue working on this and apply for further funds in the coming years. This will be aimed in our future plans to complete the precise mechanisms underlying these effects. At this stage, reporting these data is very interesting. We hope the reviewer accepts this.
Minor:
Comment: Line 86, “Adult male Wistar male rats ” should be “Adult Wistar male rats”, “Adult male Wistar rats”, or “Male adult Wistar rats”.
Response: Corrected.
Comment: Line 94, “purchased from (IPEX Corp, Canada)” should be “purchased from IPEX Corp (Canada)” .
Response: Corrected.
Comment: In 2.3 section, control rats: daily treated with 250 µl of 0.9% normal saline ; 2) AuNPs-treated rats: daily treated with 50 L of 10 nm AuNPs ; 3) DOX-treated rats: treated with an accumulative dose of DOX solution (15 mg/kg) and co-treated with 250 µl of 0.09% normal saline 4) DOX + AuNPs-treated rats: treated with DOX (15 mg/kg) and co-treated daily with 250 µl of 10 nm AuNPs. In a standard design, the volume of each group should be the same, however in the present study the authors did not follow this principle.
Response: Sorry for these mistakes, all volumes were unified. These are printing errors and corrected
Comment: In 2.4 section, I don’t think it is necessary to discuss the “Dose selection” in this part. I suggest the authors combine section 2.3 and 2.4, or discuss “Dose selection” in the discussion section.
Response: The reviewer is correct. Both sessions are combined together now.
Comment: Line 119, “All rats were authenticated by cervical dislocation” . If I have understood correctly, “authenticated” should be “sacrificed”.
Response: Corrected as suggested
Comment: In section 2.5, the authors described how to collect tissue and serum samples which would be used later. However, they missed to introduce how to get plasma.
Response: Thank you for this observation, this has been corrected and added.
Comment: Line 166, “Also, were all provided by ...” This sentence did not have subject.
Response: Corrected.
Comment: Line 181, “(Al-Qahtani et al., 2022)” should be deleted and “briefly” should be “Briefly”.
Response: Both were corrected.
Comment: Line 1987, “(Alsham- 197 mari et al., 2021)” should be deleted.
Response: Corrected
Comment: Panels of each figure should be correctly numbered, e.g.: Figures 2 and 4.
Response: Both figures were corrected. In addition, Figure 3 was labeled.
Comment: Lines 365-366, the sentence is incomplete.
Response: Corrected.
Reviewer 2 Report
Gold nanoparticles are infused with Doxorubicin-induced transport, so it is meaningful to nanoparticles the damage caused by gold nanoparticles. This paper needs to state whether the Doxorubicin dose used in this study is consistent with the dose commonly used. In this paper, the basis for the selection of time and dosage needs to be elaborated clearly! In this paper, the effects of many targets, such as inflammation, antioxidant, apoptosis and other pathways, need a main line to link them together. In addition, the following issues need to be revised.
1. 15 mg/kg; 3 mg/week: What does that amount translate to for human use? Is it the dose commonly used by the population?
2. Table 4 SOD values, the arrangement is not neat!
3. In Figure 1, mRNA expression levels of some control groups were not 1? Most graphs have this problem.
4. Figure 1 is divided into four graphs, each labeled with an ABCD.
5. In Figure 2, the expression of PPARα in mRNA and protein is inconsistent. Please explain why.
6. Please mark p65 clearly on the right of the grayscale in Figure 4. In addition, why did the author not consider the determination of phosphorylated p65?
7. In Figure 5, please use different colored arrows to make it easier to distinguish. In addition, from the perspective of logical structure, HE staining images should be placed in front of mRNA and protein determination.
8. Lines 404-406 have white space.
9. Line 456 nanny or many?
10. Line 477 has an extra space
11. The discussion part is miscellaneous and needs to be simplified. It is suggested that the author illustrate the mechanism of the corresponding signal pathway with graphs and set up subheadings to make it more logical and hierarchical.
Author Response
Reviewer 2
Gold nanoparticles are infused with Doxorubicin-induced transport, so it is meaningful to nanoparticles the damage caused by gold nanoparticles. This paper needs to state whether the Doxorubicin dose used in this study is consistent with the dose commonly used. In this paper, the basis for the selection of time and dosage needs to be elaborated clearly! In this paper, the effects of many targets, such as inflammation, antioxidant, apoptosis and other pathways, need a main line to link them together. In addition, the following issues need to be revised.
Comment: 15 mg/kg; 3 mg/week: What does that amount translate to for human use? Is it the dose commonly used by the population?
Response: several doses of DOX has been used in human based on body surface area. These are 60-75 mg/m² over a period of 21Days OR 60 mg/m²/14Days 40-60 mg/m²/21-28 Days or 20 mg/m²/dose/week. To convert our dose (mg/kg) in rats to human dose in mg/m2 , it should be multiplied by Km factor =6. This should be 3x 6 = 18 mg/m2/week and 60 mg/m2/21 dats which is very close to the human dose. In addition, we have added many references to support this dose in rats to induce liver damage. For dose conversion between various species, including rats and humans, the reviewer can refer to these articles
1) Nair AB, Jacob S. A simple practice guide for dose conversion between animals and human. J Basic Clin Pharm. 2016;7(2):27-31. doi:10.4103/0976-0105.177703
2) Mohamed J. Saadh*1, Mansour Haddad2, Moeen F. Dababneh1, Mohammad F. Bayan2, Bilal A. Al-Jaid. A Guide for Estimating the Maximum Safe Starting Dose and Conversion it between Animals and Humans. Sys Rev Pharm 2020;11(8):98-101
Comment: Table 4 SOD values, the arrangement is not neat!
Response: Corrected
Comment: In Figure 1, mRNA expression levels of some control groups were not 1? Most graphs have this problem.
Response: Yes, that is corrected because the mRAN of all groups even the control group, were normalized to the expression of actin of the same group. So the presentation shows the expression of the target gene relative to its own action. To present it as 1, then the calculation should be changed, and the expression of target genes (y-axis) should be performe as % of control. So, our way of presentation is correct and valid.
Comment: Figure 1 is divided into four graphs, each labeled with an ABCD.
Response: Corrected
Comment: In Figure 2, the expression of PPARα in mRNA and protein is inconsistent. Please explain why.
Response: Thank you for this nice comment, we have explained this in the result section, which will help the reader to understand the data more clearly.
Comment: Please mark p65 clearly on the right of the grayscale in Figure 4. In addition, why did the author not consider the determination of phosphorylated p65?
Response: This was corrected. In this study, we just provided evidence that AuNPs act by exaggerating hepatic steatosis and hyperlipidemia in the livers of DOX-treated rats the oxidative and inflammatory response, and through modulating Nrf2 and NF-kb. This highlight the regulatory role of these AUNPs on these pathways, and this was supported by the qPCR, western blot, and biochemical measurements of these factors. Regarding measuring p-NF-KB, as the reviewer knows, the regulation of NF-Kb is very complicated and depends on numerous upstream regulators, such as IKK, AMPK SIRT1, which can phosphorylate or deacetylate this transcription factor. So, to figure out the upstream mechanism, we need to study the expression of AMPK and SIRT1, as well as other regulatory proteins, as well as the phosphorylation and deacetylation rates of both factors. This will be a lot of work which requires months of work and need a special budget, which we don’t have at this time. We need to continue working on this and apply for further funds in the coming years. This will be aimed in our future plans to complete the precise mechanisms underlying these effects. At this stage, reporting these data is very interesting. We hope the reviewer accepts this
Comment: In Figure 5, please use different colored arrows to make it easier to distinguish. In addition, from the perspective of logical structure, HE staining images should be placed in front of mRNA and protein determination.
Response: Corrected
Comment: Lines 404-406 have white space.
Response: Corrected.
Comment: Line 456 nanny or many?
Response: Corrected.
Comment: Line 477 has an extra space
Comment: The discussion part is miscellaneous and needs to be simplified. It is suggested that the author illustrate the mechanism of the corresponding signal pathway with graphs and set up subheadings to make it more logical and hierarchical.
Response: We have revised the discussion part and rephrased it. All essential information needed to explain the findings of this study is explained in the current new discussion. In addition, we have added a graphical abstract to show the mechanistic effect of the drugs of this study and make it more readable to the reader.
Round 2
Reviewer 2 Report
In order to better present the results, the manuscript needs continuous improvement. I mentioned last time, the author did not carefully revise the draft, please revise and improve it this time.
1. For example, the mRNA expression level of part of the control group is not 1, which needs to be changed. This is the result presented by most published high-quality papers. The authors can search the tea query by themselves. Find some references.
2. Please use different colored arrows to make it easier to distinguish the tags, and the author has not changed.
3. Different groups in the figure use different colors to improve the readability of the article.
4. The arrows in Figure 6 are crossed.
Author Response
Reviewer 2
In order to better present the results, the manuscript needs continuous improvement. I mentioned last time the author did not carefully revise the draft, please revise and improve it this time.
General response: The comments issued by the reviewer were very interesting and important to improve the quality of the paper. All these comments were considered and corrected. As a result, they really resulted in beautiful data presentation
- For example, the mRNA expression level of part of the control group is not 1, which needs to be changed. This is the result presented by most published high-quality papers. The authors can search the tea query by themselves. Find some references.
Response: The reviewer is correct, and we have corrected the figures as folds of control
- Please use different colored arrows to make it easier to distinguish the tags, and the author has not changed.
Response: Different colors were used. Very beautiful, thank you
- Different groups in the figure use different colors to improve the readability of the article.
Response: Different colors were used as suggested for each group.
- The arrows in Figure 6 are crossed.
- Response: Corrected.